# LATENT CONCEPT-BASED EXPLANATION OF NLP MODELS

## ABSTRACT

Interpreting and understanding the predictions made by deep learning models poses a formidable challenge due to their inherently opaque nature. Many previous efforts aimed at explaining these predictions rely on input features—specifically, the words within NLP models. However, such explanations are often less informative due to the discrete nature of these words and their lack of contextual verbosity. To address this limitation, we introduce the Latent Concept Attribution method (`LACOAT`), which generates explanations for predictions based on latent concepts. Our founding intuition is that a word can exhibit multiple facets, contingent upon the context in which it is used. Therefore, given a word in context, the latent space derived from our training process reflects a specific facet of that word. `LACOAT` functions by mapping the representations of salient input words into the training latent space, allowing it to provide predictions with context-based explanations within this latent space. We will make the code of `LACOAT` available to the research community.

## 1 INTRODUCTION

The opaqueness of deep neural network (DNN) models is a major challenge to ensuring a safe and trustworthy AI system. Extensive and diverse research works have attempted to interpret and explain these models. One major line of work strives to understand and explain the prediction of a neural network model using attribution of input features to prediction (Sundararajan et al., 2017b; Denil et al., 2014). These input features are words in the text domain. One limitation of explanation based on input words is its discrete nature and lack of contextual verbosity. A word consists of multifaceted aspects such as semantic, morphological, and syntactic roles in a sentence. Consider the example of the word "trump" in Figure 1. It has several facets such as a verb, a verb with specific semantics, a named entity and a named entity representing a particular aspect such as tower names, presidents, family names, etc. We argue that given various contexts of a word in the training data, the model learns these diverse facets during training. Given an input, depending on the context a word appears, the model uses a particular facet of the input words in making the prediction. Circling back to the input feature based explanation, the explanation based on salient words alone does not reflect the facets of the word the model has used in the prediction and results in a less informed explanation. On the contrary, an explanation enriched with facets of a salient word is more insightful than the salient word alone and may additionally highlight potential issues in the training of the model.

Dalvi et al. (2022) shows that the latent space of DNNs represents the multifaceted aspects of words learned during training. The clustering of training data contextualized representations provides access to these multifaceted concepts, later referred to as *latent concepts*. Given an input word in context at test time, we hypothesize that the alignment of its contextualized representation to a latent concept represents the facet of the word being used by the model for that particular input. We further hypothesize that this latent concept serves as a correct and enriched explanation of the input word. To this end, we propose the LAtent COncept ATtribution (`LACOAT`) method that generates an explanation of a model's prediction using the latent concepts. `LACOAT` discovers latent concepts of every layer of the model by clustering high-dimensional contextualized representations of words in the training corpus. Given a test instance, it identifies the most salient input representations of every layer with respect to the prediction and dynamically maps them to the latent concepts of the training data. The shortlisted latent concepts serve as an explanation of the prediction. Lastly, `LACOAT` in-

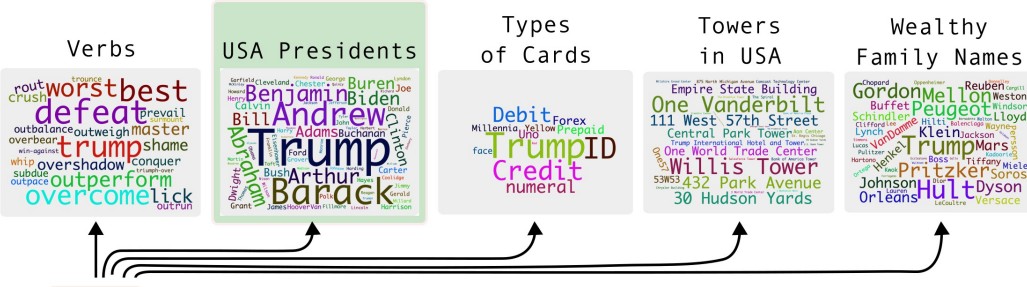

Figure 1: An example of various facets of word "trump"

tegrates a plausibility module that takes the latent concept-based explanation as input and generates a human-friendly explanation.

LACOAT is a local explanation method that provides an explanation of a single test instance. The reliance on the training data latent space makes the explanation reliable and further reflects on the quality of learning of the model and the training data. We perform qualitative and quantitative evaluation of LACOAT using the part-of-speech (POS) tagging and sentiment classification tasks across three pre-trained models. LACOAT generates an enriched explanation of predictions that is useful in understanding the reason for a prediction. It also helps in understanding how the model has structured the knowledge of a task. LACOAT enables human-in-loop in the decision-making process and augments the user with information about the prediction.

## 2 METHODOLOGY

LACOAT consists of the following four modules:

- The first module, ConceptDiscoverer, discovers latent concepts of a model given a corpus.
- PredictionAttributor, the second module, selects the most salient words (along with their contextual representations) in a sentence with respect to the model's prediction.
- Thirdly, ConceptMapper, maps the representations of the salient words to the latent concepts discovered by ConceptDiscoverer and provides a latent concept-based explanation.
- PlausiFyer takes the latent concept-based explanation as input and generates a plausible and human-understandable explanation of the prediction.

Consider a sentiment classification dataset and a sentiment classification model as an example. LACOAT works as follows: ConceptDiscoverer takes the training dataset and the model as input and outputs latent concepts of the model. At test time, given an input instance, PredictionAttributor identifies the most salient input representations with respect to the prediction. ConceptMapper maps these salient input representations to the most probable latent concepts and provides them as an explanation of the prediction. PlausiFyer takes the input test sentence and its concept-based explanation and generates a human-friendly and insightful explanation of the prediction.

In the following we describe each of these modules in detail. Consider $\mathbb{M}$ represents the DNN model being interpreted, with $L$ layers, each of size $H$. $\overrightarrow{z}_{w_i}$ *contextual representation* of a word $w_i$ in an input sentence $\{w_1, w_2, ..., w_i, ....\}$. The representation can belong to any particular layer in the model, and LACOAT will generate explanations with respect to that layer.

### 2.1 CONCEPTDISCOVERER

The words are grouped together in the high-dimensional space based on various latent relations such as semantic, morphology and syntax (Mikolov et al., 2013; Reif et al., 2019). With the inclusion of

context i.e. contextualized representations, these grouping evolves into dynamically formed clusters representing a unique facet of the words called *latent concept* (Dalvi et al., 2022). Figure 1 shows a few examples of latent concepts that capture different facets of the word "trump".

The goal of `ConceptDiscoverer` is to discover latent concepts given a model $\mathbb{M}$ and a dataset $\mathbb{D}$. We follow an identical procedure to Dalvi et al. (2022) to discover latent concepts. Specifically, for every word $w_i$ in $\mathbb{D}$, we extract contextual representations $\overrightarrow{z}_{w_i}$. We then cluster these contextualized representations using agglomerative hierarchical clustering (Gowda & Krishna, 1978). Specifically, the distance between any two representations is computed using the squared Euclidean distance, and Ward's minimum variance criterion is used to minimize total within-cluster variance. The algorithm has a hyperparamter $K$ that defines the number of clusters. We optimize $K$ for each dataset as suggested by Dalvi et al. (2022). Each cluster represents a latent concept. Let $\mathcal{C} = C_1, C_2, ..., C_K$ represents the set of latent concepts extracted by `ConceptDiscoverer`, where each $C_i = w_1, w_2, ...$ is a set of words in a particular context. For sentence classification tasks, we also consider the `[CLS]` token (or a model's representative classification token) from each sentence in the dataset as a "word" and discover the latent concepts. In this case, a latent concept may consist of words only, a mix of words and `[CLS]` tokens, and `[CLS]` tokens only.

## 2.2 SALIENT REPRESENTATIONS EXTRACTION

The goal of `PredictionAttributor` is to extract salient contextual representations for a prediction $p$ from model $\mathbb{M}$ for some given input. We consider two strategies to achieve this goal:

**Position Attribution**   This strategy uses the position of the output head as an indication of the most salient contextual representation. For instance,

- In the case of sequence classification, the representation of the `[CLS]` token, $\overrightarrow{z}_{\texttt{[CLS]}}$ (or a model's representative classification token) will be considered as the most salient representation.

- In the case of masked token prediction, the representation of the `[MASK]` token ($\overrightarrow{z}_{\texttt{MASK}}$) will be considered as the most salient for making the prediction.

- In the case of sequence labeling, the representation at the time step of the prediction will be used. For example, in the case of POS tagging, for the prediction of a tag of the word `love` in the sentence `I [love] soccer`, the second time step's representations ($\overrightarrow{z}_{w_2}$) will be used.

**Saliency based Attribution**   Gradient-based methods have been effectively used to compute the saliency of the input features for the given prediction, such as pure Gradient (Simonyan et al., 2014), Input x Gradient (Shrikumar et al., 2017) and Integrated Gradients (IG) (Sundararajan et al., 2017a). For a given input $s$ and prediction $p$, gradient-based methods give attribution scores for each token in the input sequence estimating their importance to the prediction. In this work, we use IG as our gradient-based method as its a well-established method from literature. However, this module of `LACOAT` is agnostic to the choice of the attribution method, and any other method that identifies salient input representations can be used while keeping the rest of the pipeline unchanged. Formally, we first use IG to get attribution scores for every token in the input $s$, and then select the top tokens that makeup 50% of the total attribution mass (similar to top-P sampling).

## 2.3 CONCEPTMAPPER

For an input sentence at test time, `PredictionAttributor` provides the salient input representations with respect to the prediction. `ConceptMapper` maps each salient representation to a latent concept $C_i$ of the training latent space. These latent concepts highlight a particular facet of the salient representations that is being used by the model and serves as an explanation of the prediction.

`ConceptMapper` uses a logistic regression classifier that maps a contextual representation $\overrightarrow{z}_{w_i}$ to one of the $K$ latent concepts. Specifically, the model is trained using the representations of words from dataset $\mathcal{D}$ that are used by `ConceptDiscoverer` as input features and the concept index (cluster id) as their label. Hence, for a concept $C_i$ and a word $w_j \in C_i$, a training instance of the classifier is the input $x = \overrightarrow{z}_{w_j}$ and the output is $y = i$. To optimize the classifier and to evaluate its performance, we split the dataset $\mathcal{D}$ into train (90%) and test (10%), and minimize the cross-entropy

loss over all the representations. `ConceptMapper` used in the `LACOAT` pipeline is trained using the full dataset $\mathcal{D}$.

## 2.4 PLAUSIFYER

`ConceptMapper` presents latent concepts as an explanation, leaving their understanding to domain experts and model users. Interpreting these concepts can be challenging due to the need for diverse knowledge, including linguistic, worldly, and geographical expertise (as seen in Figure 1). `PlausiFyer` simplifies the interpretation of latent concepts by offering a user-friendly summary and explanation of both the latent concept given the input sentence. Mousi et al. (2023) found ChatGPT's explanations of latent concepts to be as good as, and often superior to, human explanations. `PlausiFyer` employs a similar approach, providing words from the latent concept, such as $w_1, w_2, ...,$ and the input sentence and using a Large Language Model (LLM) like ChatGPT to explain their relationship.

We use the following prompt for sentence classification tasks:

```
Do you find any common semantic, structural, lexical and topical relation
between these sentences with the main sentence? Give a more specific and
concise summary about the most prominent relation among these sentences.

main sentence: {sentence}

{sentences}

No talk, just go.
```

and the following prompt for sequence labeling tasks:

```
Do you find any common semantic, structural, lexical and topical relation
between the word highlighted in the sentence (enclosed in [[ ]]) and the
following list of words? Give a more specific and concise summary about
the most prominent relation among these words.

Sentence: {sentence}

List of words: {words}

Answer concisely and to the point.
```

We did not provide the actual prediction of the model, or the gold label to avoid biasing the explanation.

## 3 EVALUATION AND RESULTS

### 3.1 PROCEDURE AND SETTINGS

**Data** We use two tasks, Parts-of-Speech (POS) Tagging and Sentiment Classification for our experiments. The former is a sequence labeling task, where every word in the input sentence is assigned a POS tag, while the latter classifies sentences into two classes representing *Positive* and *Negative* sentiment. We use the Penn TreeBank dataset (Marcus et al., 1993) for POS Tagging and the ERASER Movie Reviews dataset (Pang & Lee, 2004; Zaidan & Eisner, 2008) for Sentiment Classification. The POS tagging dataset consists of 36k, 1.8k and 1.9k splits for train, dev and test respectively and 44 classes.

The ERASER movie review dataset consists of labeled paragraphs with human annotations of the words and phrases. We filter sentences that have a word/phrase labeled with sentiment and create a sentence-level sentiment classification dataset. The final dataset contained 13k, 1.5k and 2.7k splits for train, dev and test respectively. The dataset including all splits consists of 9.4k positive and 8.6k negative instances.

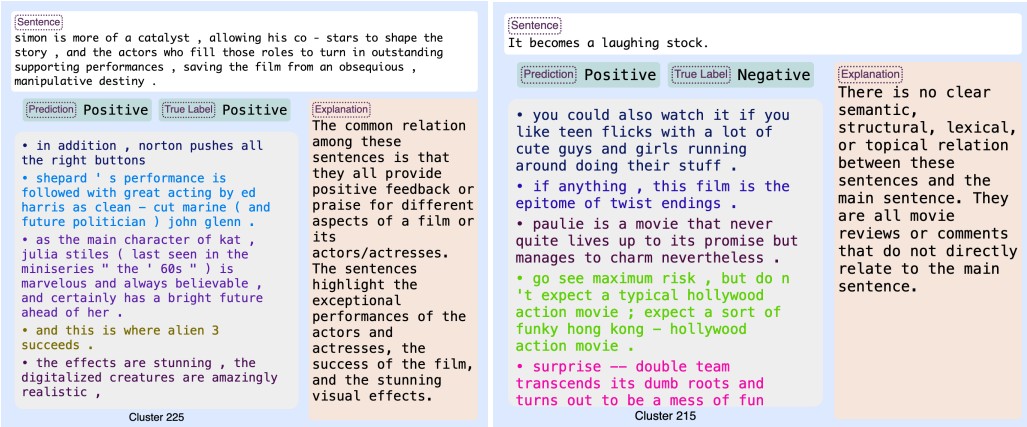

(a) A positive labeled test instance corrected predicted by the model.

(b) A negatively labeled test instance that is incorrectly predicted as positive.

Figure 2: ERASER explanation examples

**Models** We fine-tune 12-layered pre-trained models; BERT-base-cased (Devlin et al., 2019), RoBERTa-base (Liu et al., 2019) and XLM-Roberta (Conneau et al., 2020) using the training datasets of the two tasks. We use *transformers* package ((Wolf et al., 2020)) with the default settings and hyperparameters. The task performance of the models is provided in Appendix Tables 3 and 4.

**Module-specific hyperparameters** When extracting the activation and/or attribution of a word, we average the respective value over the word's subword units. We optimize number of clusters $K$ for each dataset as suggested by (Dalvi et al., 2022). We use $K = 600$ (POS tagging) and $K = 400$ (Sentiment Classification) for ConceptDiscoverer.

Since the number of words in $\mathcal{D}$ can be very high, and the clustering algorithm is limited by the total number of representations it can efficiently cluster, we filter out words with frequencies less than 5 and randomly select 15 contextual occurrences of every word with the assumption that a word may have a maximum of 15 facets. These settings are in line with Dalvi et al. (2022). In the case of [CLS] tokens, we keep all of the instances.

We use a Zero-vector as the baseline vector in PredictionAttributor's IG method, use 500 approximation steps. For ConceptMapper, we use the cross-entropy loss with L2 regularization and train the classifier with 'lbfgs' solver and 100 maximum iterations. Finally, for PlausiFyer, we use ChatGPT as the LLM with a temperature of 0 and a top_p value of 0.95.

**Layer** LACOAT works independently for every layer and generates explanations with respect to an input layer. Since the last layer is closest to the output, we found their explanations to be most useful in understanding the prediction. Below, we present the results of LACOAT using the last layer.

### 3.2 QUALITATIVE EVALUATION

Figures 2 and 3 show LACOAT's output for both tasks using layer 12 of the BERT model. The *sentence* is the input sentence, *prediction* is the output of the model, *true label* is the gold label. The *explanation* is the final output of LACOAT. *Cluster X* is the latent concept aligned with the most salient word representation at the 12th layer and X is the cluster ID. For the sentiment classification task, we discovered various [CLS] only clusters at the 12-layer. In such cases, we randomly pick five [CLS] instances from the latent concept and show their corresponding sentences in the figure.

**Correct predicted label with correct gold label** Figures 2a and 3a present a case of correct prediction with latent-concept explanation and human-friendly explanation. The latent concept-based explanations are harder to interpret especially in the case of sentence-level latent concepts as in Figure 2a compared to latent concepts consisting of words (Figure 3a). However, in both cases, PlausiFyer highlights additional information about the relation between the latent concept and

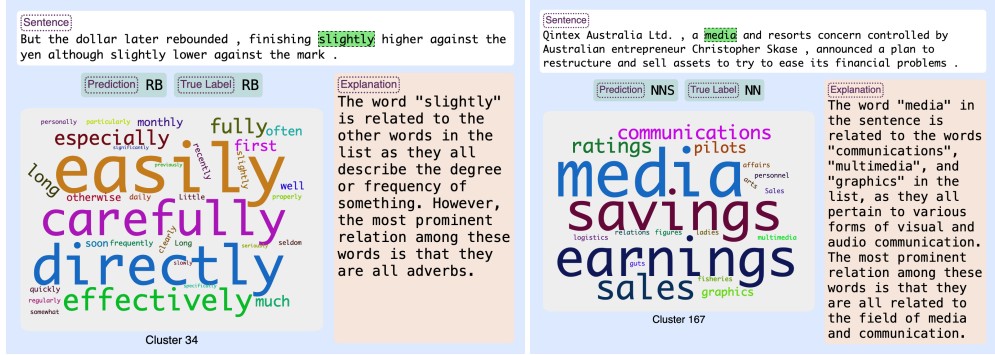

(a) An adverb with semantics showing degree and intensity of an action

(b) An incorrect prediction that can be detected from the latent concept

Figure 3: POS tagging explanation examples

the input sentence. For example, it captures that the adverbs in Figure 3a have common semantics of showing degree or frequency. Similarly, it highlights that the reason of positive sentiment in 2a is due to praising different aspects of a film and its actors and actresses.

**Wrong predicted label with correct gold label**    Figures 2b and 3b show rather interesting scenarios where the predicted label is wrong. In the case of the sentiment classification (Figure 2b), the input sentence has a negative sentiment but the model predicted it as positive. The instances of latent concepts show sentences with mixed sentiments such as "manages to charm" and "epitome of twist endings" is positive, and "mess of fun" is negative. This provides the domain expert an evidence of a possible wrong prediction. The `PlausiFyer`'s *explanation* is even more helpful as it clearly states that "there is no clear ... relation between these sentences ...". Similarly, in the case of the POS example (Figures 3b), while the prediction is Noun, the majority of words in the latent concepts are plural Nouns, giving evidence of a possibly wrong prediction. In addition, the *explanation* did not capture any morphological relationship between the concept and the input word.

In order to study how the explanation would change if it is a correct prediction, we employ the TextAttack tool (Morris et al., 2020) to create an adversarial example of the sentence in Figure 2b that flips its prediction. The new sentence replaces 'laughing' with 'kidding' which has a similar meaning but flipped the prediction to a correct prediction. Figure 6 in the appendix shows the full explanation of the augmented sentence. With the correct prediction, the latent concept changed and the *explanation* clearly expresses a negative sentiment "... all express negative opinions and criticisms ..." compared to the explanation of the wrongly predicted sentence.

**Cross model analysis**    `LACOAT` provides an opportunity to compare various models in terms of how they learned and structured the knowledge of a task. Figure 4 compares the explanation of RoBERTa (left) and XLMR (left) for identical input. Both models predicted the correct label. However, their latent concept based explanation is substantially different. RoBERTa's explanation shows a large and diverse concept where many words are related to finance and economics. The XLMR's latent concept is rather a small focused concept where the majority of tokens are units of measurement. It is worth noting that both models are fine-tuned on identical data.

## 3.3 VALIDATING THE METHODOLOGY

The correctness of `LACOAT` depends on the performance of each module it comprised off. The ideal way to evaluate the efficacy of these modules is to consider gold annotations. However, the ground truth annotations are not available for any module. To mitigate this limitation, we design various constrained scenarios where certain assumptions can be made about the representations of the model. For example, the POS tagging model optimizes POS tags so it is highly probable that the last layer representations form latent concepts that are a good representation of POS tags as suggested by various previous works (Kovaleva et al., 2019; Durrani et al., 2022). One can assume that for `ConceptDiscoverer`, the last layer latent concepts will form groupings of words based

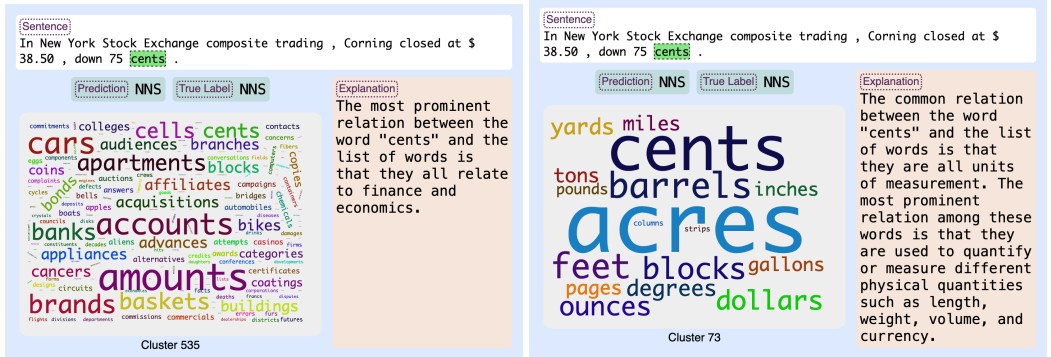

Figure 4: Comparing explanation of RoBERTa (left) and XLMR (right)

on specific tags and for `PredictionAttributor`, the input word at the position of the predicted tag should reside in a latent concept that is dominated by the words with identical tag. We evaluate the correctness of these assumptions in the following subsections.

**Latent Concept Annotation** For the sake of evaluation, we annotated the latent concepts automatically using the class labels of each task. Given a latent concept, we annotate it with a certain class if more than 90% of the words in the latent concept belong to that class. In the case of POS tagging, the latent concepts will be labeled with one of the 44 tags. In the case of ERASER, the class labels, *Positive* and *Negative*, are at sentence level. We tag a latent concept *Positive/Negative* if 90% of its tokens whether [CLS] or words belong to sentences labeled as *Positive/Negative* in the training data. The latent concepts that do not fulfill the criteria of 90% for any class are annotated as *Mixed*.

### 3.3.1 CONCEPTDISCOVERER

**A latent concept is a true reflection of the properties that a representation possesses.** `ConceptDiscoverer` identifies latent concepts by clustering the representation in the high dimensional space. We questioned whether the discovered latent concepts are a true reflection of the properties that a representation possesses. Using `ConceptDiscoverer`, we form latent concepts of the last layer and automatically annotate them as described above. We found 87%, 83% and 86% of the latent concepts of BERT, RoBERTa and XLMR that perfectly map to a POS tag respectively. We further analyzed other concepts where 95% of the words did not belong to a single tag. We found them to be of a compositional nature i.e. a concept consisting of related semantics like a mix of adjectives and proper nouns about countries such as Sweden, Swedish (Appendix Figure 5). For ERASER, we found 78%, 95% and 94% of the latent concepts of BERT, RoBERTa and XLMR to consist of either Positive or Negative sentences. The high number of class-based clusters of RoBERTa and XLMR show that at 12th layer, majority of their latent space is separated based on these two classes. Appendix Table 10 presents these figures for each layer.

### 3.3.2 PREDICTIONATTRIBUTOR

**The salient input representation correctly represents the latent space of the output.** Here, we aim to evaluate the `PredictionAttributor` module. We consider it to be correct if at least for the last (few) layer(s), the salient representation aligns with the latent concept that is dominated by the words/sentences of the same tag/class as the label. There are two ways to select a salient input representation – Position based and Saliency based. We evaluate the former as the number of times an input representation at the position of the output head maps to the latent concept that is annotated with an identical label as the output. For example, consider that the model predicts Proper Noun (PN) for the input word "Trump". In order for the input representation of the predicted label to be aligned with the latent concept, the representation of the word "Trump" on at least the last layer should be in a cluster of words whose label is PN.[1] Similarly for sentiment classification, we expect the [CLS] representation on the last layer to map to a latent concept that is dominated by the same class as the prediction. For the saliency based method, we calculate the number of times the

---

[1]We labeled concepts with a tag if 90% of the words in the concept belongs to one class.

Table 1: Accuracy of `PredictionAttributor` in mapping a representation to the correct latent concept. See Appendix Table 7, 8, 9 for full results.

| | POS | | | ERASER | | | | | |
| | Position/Saliency | | | Position based | | | Saliency based | | |
| Layers | BERT | RoBERTa | XLMR | BERT | RoBERTa | XLMR | BERT | RoBERTa | XLMR |
|---|---|---|---|---|---|---|---|---|---|
| 9 | 92.38 | 86.97 | 91.97 | 37.09 | 98.45 | 0 | 31.94 | 99.59 | 32.63 |
| 10 | 92.79 | 89.64 | 92.64 | 99.55 | 99.14 | 0 | 99.57 | 99.69 | 92.06 |
| 11 | 93.39 | 89.95 | 92.59 | 99.82 | 99.27 | 99.17 | 99.71 | 99.48 | 94.97 |
| 12 | 93.95 | 90.04 | 93.13 | 99.25 | 99.27 | 99.08 | 99.25 | 99.27 | 99.08 |

Table 2: Top 1,2 and 5 accuracy of `ConceptMapper` in mapping a representation to the correct latent concept. See Table 5, 6 in the Appendix for results on all layers. Model: BERT

| | Layers | 0 | 1 | 2 | 5 | 6 | 7 | 10 | 11 | 12 |
|---|---|---|---|---|---|---|---|---|---|---|
| POS | Top 1 | 100 | 100 | 100 | 99.03 | 97.76 | 96.51 | 92.67 | 90.86 | 84.19 |
| | Top 2 | 100 | 100 | 100 | 99.75 | 99.34 | 98.91 | 97.89 | 97.34 | 94.15 |
| | Top 5 | 100 | 100 | 100 | 99.94 | 99.83 | 99.68 | 99.68 | 99.64 | 99.05 |
| ERASER | Top 1 | 100 | 100 | 100 | 97.19 | 96.44 | 94.86 | 83.09 | 76.84 | 68.24 |
| | Top 2 | 100 | 100 | 100 | 99.63 | 99.3 | 98.97 | 92.67 | 88.02 | 83.24 |
| | Top 5 | 100 | 100 | 100 | 99.94 | 99.89 | 99.9 | 97.75 | 96.01 | 94.24 |

representation of the most salient word/[CLS] token maps to the latent concept of the identical label as that of the prediction.

We do not include `ConceptMapper` when evaluating `PredictionAttributor` and conduct this experiment using the training data only where we already know the alignment of a salient representation and the latent concept. Table 1 shows the results across the last four layers (See Appendix Table 7, 8, 9 for full results). For POS, the salient representation is identical for both the position based and saliency based methods and results in the same performance. We observed a successful match of 93.95%, 90.04% and 93.13% for BERT, RoBERTa and XLMR models respectively. We observed the mismatched cases and found them to be of compositional nature i.e. latent concepts comprised of semantically related words (see Appendix Figure 5 for examples).

For ERASER, more than 99% of the time, the last layer's salient representation maps to the predicted class label, confirming the correctness of `PredictionAttributor`. For lower layers, the performance drops and even reaches zero as in the case of XLMR. This is mainly due to the absence of class-based latent concepts in the lower layers i.e. concepts that comprised more than 90% of the tokens belonging to sentences of one of the classes. The other reason is the position-based method which fails to find the right latent concept when the most attributed word is different from the position of the output head.

### 3.3.3 CONCEPTMAPPER

**`ConceptMapper` correctly maps a new representation to the latent space.** Here, we evaluate the correctness of `ConceptMapper` in mapping a test representation to the training data latent concepts. `ConceptMapper` trains using representations and their cluster ids as labels. For every layer, we randomly split this training data into 90% train and 10% test data. Here, the test data serves as the gold standard annotation of latent concepts. We train `ConceptMapper` using the training instances and measure the accuracy of the test instances. Table 2 presents the accuracy of the POS and ERASER tasks using BERT (See Appendix Tables 5, 6 for results of other models). Top-1, Top-2 and Top-5 refer to top 1, 2 and 5 predictions of the mapper. Observing Top-1, the performance of `ConceptMapper` starts high (100%) for lower layers and drops to 84.19 and 68.24% for the last layer. We found that the latent space becomes dense on the last layer. This is in line with Ethayarajh (2019) who showed that the representations of higher layers form a cone. This causes several similar concepts close in the space. If true, the correct label should be among the top predictions of the mapper. We empirically tested it by considering the top two and top five predictions of the mapper. It achieved a performance of up to 99.05% and 94.24% for POS and ERASER respectively.

### 3.3.4 PLAUSIFYER

**The human-friendly explanation is faithful to the latent-concept based explanation** i.e. it conveys the correct relation based on which representations in a latent concept are grouped together. We rely on the findings of (Mousi et al., 2023) who showed that the annotations of latent concepts produced by generative models are better than human annotations. In our qualitative evaluation, we found rare cases where PlausiFyer was unable to capture the relation between the latent concept and the input (Appendix Figure 7 shows an example).

## 4 RELATED WORK

The explainability methods can be approached by local explanations and global explanations (Madsen et al., 2023; Sundararajan et al., 2017b; Denil et al., 2014; Selvaraju et al., 2020; Kapishnikov et al., 2021; Zhao & Aletras, 2023; Kim et al., 2018; Ghorbani et al., 2019; Jourdan et al., 2023; Zhao et al., 2023; Ribeiro et al., 2016). Lyu et al. (2023) provides a comprehensive survey on explainability methods in NLP. LACOAT is a local explanation method providing post-hoc explanations for each single instance. One of the most common ways for local explanations is to interpret the model prediction based on the input features. However, the shortcoming of this type of method is the lack of contextual verbosity, which could not interpret the multifaceted roles of the input features and could not demonstrate how the models learn the contextual and task knowledge. To solve this issue, the concept-based explanation is a popular method to identify the high-level influential factor and to have a clearer and more comprehensive understanding of the model's prediction process (Kim et al., 2018; Ghorbani et al., 2019; Zhao et al., 2023; Jourdan et al., 2023). For instance, TCAV is a global explanation method that utilizes directional derivatives to measure the model's prediction sensitivity towards a human-defined concept to generate explanations (Kim et al., 2018). A limitation of the methods is their reliance on human pre-defined concepts, which may be subject to human bias, and the concept may not represent the way the model has learned the knowledge of a task.

A number of works attempted to explain and interpret NLP models using high-level concepts extracted from hidden representations (Zhao et al., 2023; Dalvi et al., 2022; Rajani et al., 2020). Zhao et al. (2023) worked on global explanation and trained a separate surrogate model to discover latent concepts based on two optimization criteria i.e. auto encoding loss to stay faithful to the original model distribution and impact of the latent concept to prediction. Different from them, we provide local explanations and we ensure the faithfulness of latent concepts to the model by extracting them directly from the hidden representation without any supervised training. Rajani et al. (2020) used k-nearest neighbors of the training data for low-confidence predictions and showed them to be useful in revealing acquired erroneous correlations, pinpointing misclassified instances, and enhancing the performance of the finetuned model. Our latent concept discovery module is similar to Dalvi et al. (2022) and is based on hierarchical clustering. Dalvi et al. (2022) proposed it to analyze how knowledge of a task is structured in the network. LACOAT extends it to how the structured knowledge is used in the prediction and provides a human-friendly explanation.

## 5 CONCLUSION AND LIMITATIONS

We presented LACOAT that provides a human-friendly explanation of a model's prediction using the training data latent concepts. We performed a thorough evaluation of each module of LACOAT. The qualitative evaluation showed that LACOAT explanations are insightful in explaining a correct prediction, in highlighting a wrong prediction and in comparing the explanations of models. LACOAT promises to engage human-in-the-loop in the decision-making process and is an essential step towards trust in AI.

A few limitations of LACOAT are: 1) while hierarchical clustering is better than nearest neighbor in discovering latent concepts as established by Dalvi et al. (2022), it has computational limitations and it can not be easily extended to a corpus of say 1M tokens. 2) LACOAT requires access to the training data and the model for explanation which may not be available for large language models such as Llama and ChatGPT.

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
