# A  APPENDIX

## A.1  POS SEQUENCE TAGGER

We tuned several transformers BERT-base-cased, RoBERTa and XLM-RoBERTa. We used standard splits for training, development and test data that we used to carry out our analysis. The splits to preprocess the data are available through git repository[2]. See Table 3 for statistics and classifier accuracy.

Table 3: The fine-tuned performance of models, data statistics (number of sentences) on training, development, and test sets used in the experiments, and the number of tags to be predicted for the POS sequence tagging task. Model: BERT, RoBERTa, XLM-R

| Task | Train | Dev | Test | Tags | BERT | RoBERTa | XLM-R |
|------|-------|-----|------|------|------|---------|-------|
| POS | 36557 | 1802 | 1963 | 48 | 96.81 | 96.70 | 96.75 |

## A.2  SENTIMENT CLASSIFICATION

Table 4: The fine-tuned performance of models, data statistics (number of sentences) on training, development, and test sets used in the experiments, and the number of tags to be predicted for the sentiment classification task. Model: BERT, RoBERTa, XLM-R

| Task | Train | Dev | Test | Tags | BERT | RoBERTa | XLM-R |
|------|-------|-----|------|------|------|---------|-------|
| ERASER | 13878 | 1516 | 2726 | 2 | 94.53 | 96.31 | 93.80 |

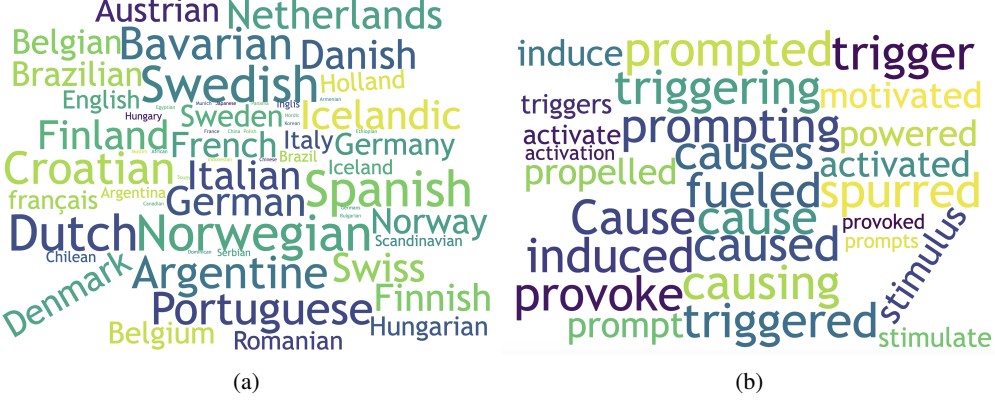

(a)                                          (b)

Figure 5: Compositional concepts: (a) A cluster representing countries (NNP) and their adjectives (JJ), (b) Different form of verbs (Gerunds, Present and Past participles). We found that the concepts are not always formed aligning to the output class. Some concepts are formed by combining words from different classes. For example in Figure 5a, the concept is composed of nouns (specifically countries) and adjectives that modify these country nouns. Similarly Figure 5b describes a concept composed of different forms of verbs.

---

[2] https://github.com/nelson-liu/contextual-repr-analysis

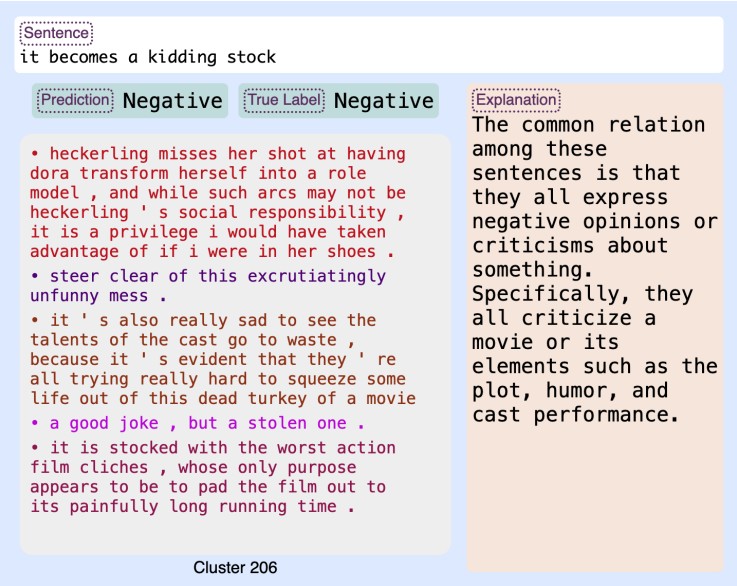

Figure 6: An augmented example for the test instance in Figures 2b: The augmented sentence replaced the 'laughing' with 'kidding' which has a similar meaning. The label of the augmented sentence becomes positive, which is matched with the gold label. The new predicted latent concept is more closely aligned with the main sentence. The model may not learn the implicit meaning of the 'laughing stock' in the sentence.

Table 5: Top 1, 2, and 5 accuracies of `ConceptMapper` in mapping a representation to the correct latent concept for the POS tagging task. The top-5 performance reaches above 99% for all models demonstrating that the correct latent concept is among the top probable latent concepts of `ConceptMapper`.

| | POS | | | | | | | | |
|---|---|---|---|---|---|---|---|---|---|
| | **BERT** | | | **RoBERTa** | | | **XLM-R** | | |
| Layer | Top-1 | Top-2 | Top-5 | Top-1 | Top-2 | Top-5 | Top-1 | Top-2 | Top-5 |
| Layer 0 | 100 | 100 | 100 | 99.91 | 99.95 | 99.98 | 99.99 | 100 | 100 |
| Layer 1 | 100 | 100 | 100 | 99.92 | 99.94 | 99.98 | 100 | 100 | 100 |
| Layer 2 | 100 | 100 | 100 | 99.76 | 99.92 | 99.98 | 99.72 | 99.98 | 100 |
| Layer 3 | 99.85 | 99.98 | 100 | 99.38 | 99.85 | 99.98 | 98.25 | 99.60 | 99.98 |
| Layer 4 | 99.72 | 99.92 | 99.97 | 98.67 | 99.58 | 99.87 | 97.72 | 99.60 | 99.98 |
| Layer 5 | 99.03 | 99.75 | 99.94 | 97.69 | 99.15 | 99.73 | 97.05 | 99.23 | 99.91 |
| Layer 6 | 97.76 | 99.34 | 99.83 | 96.52 | 98.71 | 99.59 | 95.8 | 98.95 | 99.76 |
| Layer 7 | 96.51 | 98.91 | 99.68 | 94.72 | 98.11 | 99.57 | 93.92 | 98.31 | 99.80 |
| Layer 8 | 95.27 | 98.52 | 99.79 | 92.56 | 97.55 | 99.52 | 94.20 | 98.52 | 99.80 |
| Layer 9 | 94.54 | 98.25 | 99.70 | 92.24 | 97.48 | 99.55 | 92.79 | 97.82 | 99.73 |
| Layer 10 | 92.67 | 97.89 | 99.68 | 91.61 | 97.19 | 99.55 | 92.03 | 97.66 | 99.60 |
| Layer 11 | 90.86 | 97.34 | 99.64 | 90.72 | 96.77 | 99.58 | 90.40 | 97.28 | 99.67 |
| Layer 12 | 84.19 | 94.15 | 99.05 | 86.88 | 95.13 | 99.15 | 85.07 | 94.57 | 99.08 |

Table 6: Top 1, 2, and 5 accuracy of `ConceptMapper` in mapping a representation to the correct latent concept for the ERASER task. The top-5 performance reaches above 90% for all models demonstrating that the correct latent concept is among the top probable latent concepts of `ConceptMapper`.

| | ERASER | | | | | | | | |
|---|---|---|---|---|---|---|---|---|---|
| | **BERT** | | | **RoBERTa** | | | **XLM-R** | | |
| Layer | Top-1 | Top-2 | Top-5 | Top-1 | Top-2 | Top-5 | Top-1 | Top-2 | Top-5 |
| 0 | 100 | 100 | 100 | 99.95 | 100 | 100 | 100 | 100 | 100 |
| 1 | 100 | 100 | 100 | 99.86 | 99.98 | 100 | 100 | 100 | 100 |
| 2 | 100 | 100 | 100 | 99.89 | 99.98 | 100 | 99.9 | 100 | 100 |
| 3 | 98.80 | 100 | 100 | 99.44 | 99.83 | 99.96 | 99.57 | 99.99 | 100 |
| 4 | 97.84 | 99.85 | 99.99 | 99.28 | 99.73 | 99.91 | 99.4 | 99.96 | 100 |
| 5 | 97.19 | 99.63 | 99.94 | 98.4 | 99.5 | 99.84 | 99.12 | 99.84 | 99.96 |
| 6 | 96.44 | 99.30 | 99.89 | 97.35 | 99.15 | 99.82 | 98.9 | 99.84 | 99.96 |
| 7 | 94.86 | 98.97 | 99.90 | 96.13 | 98.74 | 99.63 | 98.22 | 99.62 | 99.9 |
| 8 | 93.26 | 97.99 | 99.67 | 87.42 | 95.14 | 98.43 | 98.13 | 99.48 | 99.84 |
| 9 | 90.42 | 96.97 | 99.20 | 75.38 | 88.14 | 96.07 | 96.37 | 98.77 | 99.66 |
| 10 | 83.09 | 92.67 | 97.75 | 65.84 | 81.13 | 93.46 | 89.12 | 95.2 | 98.61 |
| 11 | 76.84 | 88.02 | 96.01 | 65.91 | 81.36 | 93.43 | 70.99 | 84.31 | 94.18 |
| 12 | 68.24 | 83.24 | 94.24 | 70.83 | 84.54 | 95.67 | 55.3 | 75.08 | 91.74 |

Table 7: Position/Saliency-based method: accuracy of `PredictionAttributor` in mapping a representation to the correct latent concept in the POS tagging task.

| | POS | | |
|---|---|---|---|
| Layer | BERT | RoBERTa | XLM-R |
| Layer 0 | 16.81 | 14.29 | 17.66 |
| Layer 1 | 17.79 | 16.49 | 18.89 |
| Layer 2 | 21.16 | 20.18 | 20.71 |
| Layer 3 | 22.79 | 20.13 | 31.03 |
| Layer 4 | 29.70 | 24.65 | 40.51 |
| Layer 5 | 46.74 | 29.26 | 60.31 |
| Layer 6 | 73.19 | 42.38 | 77.32 |
| Layer 7 | 84.52 | 57.46 | 85.78 |
| Layer 8 | 90.68 | 82.84 | 89.41 |
| Layer 9 | 92.38 | 86.97 | 91.97 |
| Layer 10 | 92.79 | 89.64 | 92.64 |
| Layer 11 | 93.39 | 89.95 | 92.59 |
| Layer 12 | 93.95 | 90.04 | 93.13 |

Table 8: Position-based method: accuracy of `PredictionAttributor` in mapping a representation to the correct latent concept in the ERASER task. The reason of zero values is that the position-based method fails to find the right latent concept when the most attributed word is different from the position of the output head.

| Layer | ERASER | | |
| | BERT | RoBERTa | XLM-R |
|---|---|---|---|
| Layer 0 | 0 | 0 | 0 |
| Layer 1 | 0 | 0 | 0 |
| Layer 2 | 0 | 0 | 0 |
| Layer 3 | 0 | 0 | 0 |
| Layer 4 | 0 | 0 | 0 |
| Layer 5 | 0 | 0 | 0 |
| Layer 6 | 0 | 0 | 0 |
| Layer 7 | 0 | 0 | 0 |
| Layer 8 | 0 | 99.11 | 0 |
| Layer 9 | 37.09 | 98.45 | 0 |
| Layer 10 | 99.55 | 99.14 | 0 |
| Layer 11 | 99.82 | 99.27 | 99.17 |
| Layer 12 | 99.25 | 99.27 | 99.08 |

Table 9: Saliency-based method: accuracy of `PredictionAttributor` in mapping a representation to the correct latent concept in the ERASER task. The reason of very low values for lower layers is mainly due to the absence of class-based latent concepts in the lower layers i.e. concepts that comprised more than 90% of the tokens belonging to sentences of one of the classes.

| Layer | ERASER | | |
| | BERT | RoBERTa | XLM-R |
|---|---|---|---|
| Layer 0 | 6.40 | 12.08 | 7.46 |
| Layer 1 | 7.12 | 12.46 | 5.57 |
| Layer 2 | 7.66 | 17.29 | 6.36 |
| Layer 3 | 7.13 | 22.00 | 8.03 |
| Layer 4 | 12.18 | 20.08 | 9.71 |
| Layer 5 | 13.24 | 24.25 | 8.88 |
| Layer 6 | 11.18 | 17.26 | 8.75 |
| Layer 7 | 12.80 | 39.87 | 14.05 |
| Layer 8 | 4.06 | 92.84 | 15.75 |
| Layer 9 | 31.94 | 99.59 | 32.63 |
| Layer 10 | 99.57 | 99.69 | 92.06 |
| Layer 11 | 99.71 | 99.48 | 94.97 |
| Layer 12 | 99.25 | 99.27 | 99.08 |

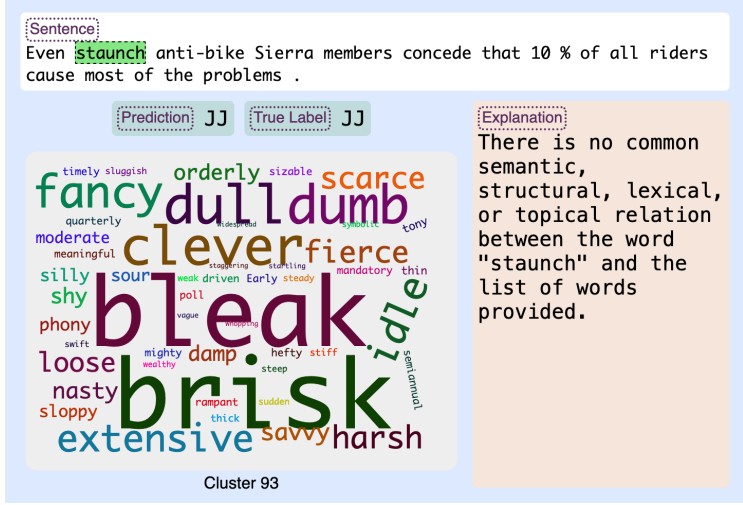

Figure 7: Example of `PlausiFyer` not working well: Both the prediction and the majority of the words in the latent concept are adjectives; however, the explanation did not capture any relationship between them.

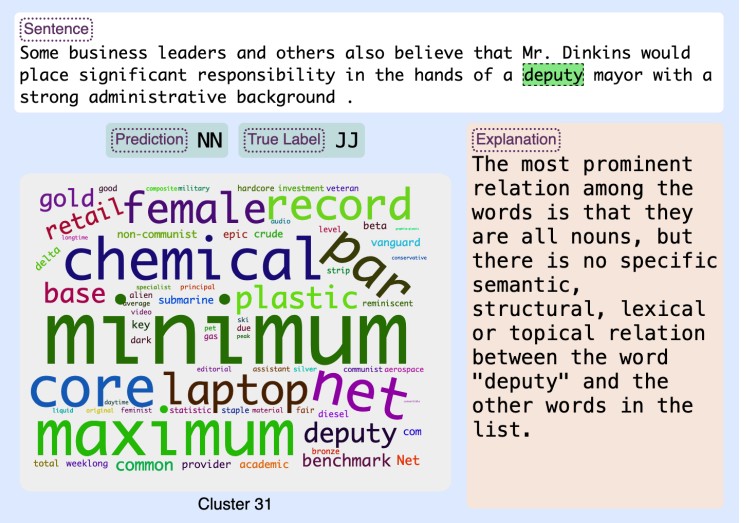

Figure 8: An incorrect prediction (noun vs adjective) based on a latent concept made up of a mixture of nouns and adjectives: the 'deputy' in this case is an adjective. The prediction aligns with a mixed cluster that contains both nouns and adjectives and the model may not learn to distinguish the 'noun' and 'adjective' in this case. The latent concept explanation is useful for the user to know that the model has used a mixed latent space for the prediction. The Explanation is rather wrong since it mentions that all these words are nouns.

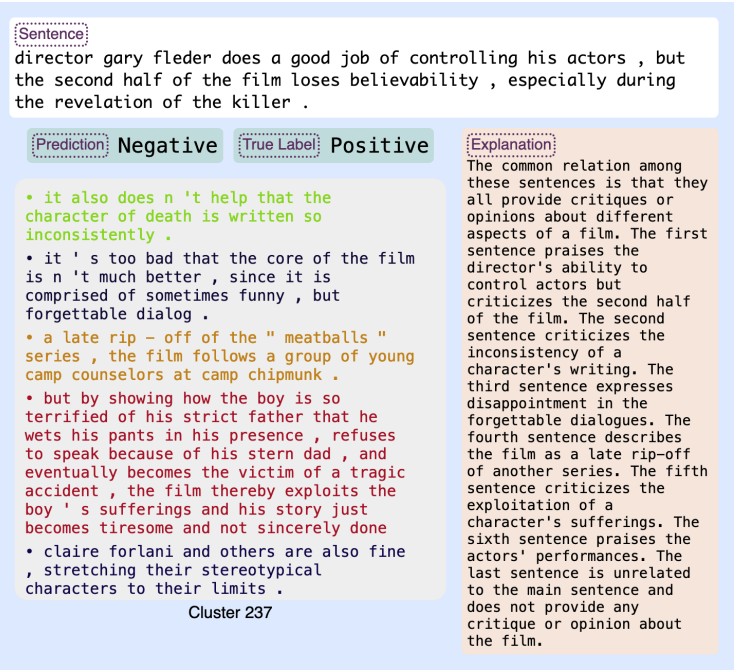

Figure 9: A correct prediction but incorrect ground truth label: The test instance emphasizes the movie's shortcomings and uses the word "especially" to highlight the flaws. The explanation is rather long but it correctly highlights that the sentences are about "critiques or opinions"

Table 10: Number of clusters for each polarity: 'Neg' for negative Label, 'Pos' for positive, and 'Mix' for mix label. Total number of clusters are 400.

| | ERASER | | | | | | | | |
| | BERT | | | RoBERTa | | | XLM-R | | |
| Layer | Neg | Pos | Mix | Neg | Pos | Mix | Neg | Pos | Mix |
|---|---|---|---|---|---|---|---|---|---|
| Layer 0 | 49 | 1 | 350 | 45 | 0 | 355 | 55 | 0 | 345 |
| Layer 1 | 53 | 1 | 346 | 50 | 0 | 350 | 58 | 0 | 342 |
| Layer 2 | 51 | 1 | 348 | 49 | 0 | 351 | 62 | 0 | 338 |
| Layer 3 | 53 | 0 | 347 | 60 | 0 | 340 | 62 | 0 | 338 |
| Layer 4 | 57 | 0 | 343 | 52 | 0 | 348 | 69 | 0 | 331 |
| Layer 5 | 56 | 0 | 344 | 51 | 0 | 349 | 68 | 0 | 332 |
| Layer 6 | 57 | 0 | 343 | 45 | 1 | 354 | 59 | 1 | 340 |
| Layer 7 | 51 | 0 | 349 | 56 | 2 | 342 | 68 | 0 | 332 |
| Layer 8 | 49 | 0 | 351 | 116 | 25 | 259 | 71 | 0 | 329 |
| Layer 9 | 66 | 4 | 330 | 226 | 126 | 48 | 82 | 7 | 311 |
| Layer 10 | 125 | 31 | 244 | 235 | 140 | 25 | 257 | 92 | 51 |
| Layer 11 | 174 | 49 | 177 | 258 | 120 | 22 | 256 | 110 | 34 |
| Layer 12 | 230 | 81 | 89 | 254 | 126 | 20 | 105 | 270 | 25 |