# OpenReview forum: "Latent Concept-based Explanation of NLP Models"
_ICLR.cc/2024/Conference — Submitted to ICLR 2024_

### Official Review · Reviewer_Zm9D · 2023-10-19

**Soundness:** 2 fair
**Presentation:** 3 good
**Contribution:** 1 poor
**Rating:** 5
**Confidence:** 2

**Summary:**

This paper attempts to explain a model's prediction using latent concepts. The proposed approach is comprised of a few loosely connected components including concept discoverer, prediction attributor, concept mapper, and plausifyer. Concept discoverer clusters words while disambiguating them based on their senses using the training data. Prediction attributor uses either a set of handcrafted rules, or integrated gradient. The experiments evaluate each component and examples are provided throughout the paper.

**Strengths:**

- S1. The paper raises an interesting question about ambiguous natural language explanation and tries to disambiguate the sense (latent concept).
- S2. The paper provides examples and experiments.

**Weaknesses:**

- W1. The components are loosely connected, which isn't necessarily a bad thing by itself. However, each component is either simplistic or an existing approach.
- W2. The evaluation is done with automatically generated labels, and in this particular case, they can be deceptive because of the last layer assumption used to generate them. Also, if they are the targets, one can just adopt the method used to generated labels to replace the proposed method.
- W3. The evaluation is limited to simplistic tasks only.

**Questions:**

- Q1: Why do you use squared Euclidean distance which is not a metric?
- Q2: How is this different from topic modeling, or many other word sense disambiguation especially as other components are mostly direct application of existing approaches?
- Q3: What are the examples of the ambiguity handled with the concepts in the evaluation? Are they apparent with the input text?

---

> ### Author Response · Authors · 2023-11-21
>
> **W: The components are loosely connected, which isn't necessarily a bad thing by itself. However, each component is either simplistic or an existing approach.**
>
> **A:** Our work combines the encoded concepts with the attribution analysis to explain the model’s decision in light of the utilized latent space. Our novel formulation allows us to combine what the model has learned or knows with what it uses for making predictions. We argue that it is not straightforward to simply combine other works since the efficacy of every module affects the final output of the model. This is evident from various module-level evaluations presented in the paper.
>
> **W: The evaluation is done with automatically generated labels, and in this particular case, they can be deceptive because of the last layer assumption used to generate them. Also, if they are the targets, one can just adopt the method used to generated labels to replace the proposed method.**
>
> **A:** The evaluation of explanations is a difficult problem. We selected the POS tagging task to design an evaluation that can measure the performance of each module involved in our methodology.
>
> **W: The evaluation is limited to simplistic tasks only.**
>
> **A:** While our approach can be applied to explain the prediction of any task, the evaluation of explanation is difficult. We have extended the explanation generation to the MNLI task as well. Moreover, we plan to perform human evaluation to quantify the usefulness of the explanation.
>
> **Q: Why do you use squared Euclidean distance which is not a metric?**
>
> **A:** Squared euclidean distance is one of the most common metrics used in hierarchical clustering and it is based on the assumption that the space is Euclidean.
>
> **Q: How is this different from topic modeling, or many other word sense disambiguation especially as other components are mostly direct application of existing approaches?**
>
> **A:** It does seem related where a latent concept can be seen as clusters of topics. Similarly, a word with a different sense will have a different latent concept. In our case, the latent concepts are derived directly from the embedding space of the underlying model, and reflect the model's knowledge about the task.
>
> **Q: What are the examples of the ambiguity handled with the concepts in the evaluation? Are they apparent with the input text?**
>
> **A:** We assume that the reviewer is referring to ambiguity related to different semantics of a word. Since we consider contextualized representations of a word, the semantic differences will cause different instances of a word to appear in the same or different latent concepts.

---

### Official Review · Reviewer_6gzN · 2023-11-01

**Soundness:** 2 fair
**Presentation:** 2 fair
**Contribution:** 2 fair
**Rating:** 5
**Confidence:** 2

**Summary:**

The authors propose a latent concept attribution method called LACOAT that goes through a set of modules and generates explanations for predictions based on latent concepts. Their method hinges on the fact that words can have different contextualized senses and they assume the latent spaces of models utilize this.

**Strengths:**

1) The motivation is sound. I understand and buy the fact that we need our models to be able to explain predictions for numerous reasons and having a way to do this via the latent concepts the model has encoded is a great way to try and do this.

**Weaknesses:**

1) I think a major weakness of the overall method is poor scalability. Clustering at scale would be quite expensive and large pretrained datasets are in the 3T total token range (e.g. Dolma) which would definitely be infeasible.

2) I think the experimentation could be extended. POS and sentiment are very small and relatively simple. The types of models that you're fine-tuning are also not very broad (no decoder only models like GPT here). What happens when you evaluate on an NLI task? Could you extend this to work on a task like question answering and evaluate that?

3) It's really hard for me to figure out the appropriate baselines here, which I think is a potential weakness. It's hard for me to contextualize the results you have.

**Questions:**

1) How would/could you modify your algorithm to work with a pretrained LM without exhaustive fine-tuning on a target task?

2) If you have some resource limitations, but wanted to scale up these experiments to a much larger model (e.g 1.5B parameter model), would this be straight forward to do? Could you fine-tune on a subset?

---

> ### Author Response · Authors · 2023-11-21
>
> **W: I think a major weakness of the overall method is poor scalability. Clustering at scale would be quite expensive and large pretrained datasets are in the 3T total token range (e.g. Dolma) which would definitely be infeasible.**
>
> **A:** Our method should be considered as a step towards providing explanations of the prediction of pretrained models and there are indeed a few limitations to it. The current scope of the approach aims at providing explanation of task specific models where latent concepts of a task can be extracted. This scope nicely covers a majority of NLP tasks. Creating latent concepts of the pretrained data is indeed impractical and we plan to look into it as a future research direction.
>
> **W: I think the experimentation could be extended. POS and sentiment are very small and relatively simple. The types of models that you're fine-tuning are also not very broad (no decoder only models like GPT here). What happens when you evaluate on an NLI task?**
>
> **A:** Our approach is model agnostic and it can easily be applied to decoder only models. We performed experiments using the MNLI task and found latent concepts at the last layer consisting of sentence pairs. The sentences are grouped together in a latent concept based on how the hypothesis relates with the premise e.g. summary, rephrasing the argument, logical relationship, etc.
>
> **W: Could you extend this to work on a task like question answering and evaluate that?
> It's really hard for me to figure out the appropriate baselines here, which I think is a potential weakness. It's hard for me to contextualize the results you have.**
>
> **A:** We extended the experiments to the NLI task for the rebuttal. The baseline and evaluation in the case of free form explanation is indeed very hard. We plan to perform a human evaluation to quantify the usefulness of our approach.
>
> **Q: How would/could you modify your algorithm to work with a pretrained LM without exhaustive fine-tuning on a target task?**
>
> **A:** We assume that the reviewer is referring to prompting LLMs to perform a task without fine-tuning. Our approach relies on the latent concepts of a task with respect to a model. Given training data of a task, say sentiment analysis, one can prompt the model using the training data, extract activations and create latent concepts of the task. At test time, given the generated token(s), one can identify latent concepts and generate an explanation of the prediction.
>
> **Q: If you have some resource limitations, but wanted to scale up these experiments to a much larger model (e.g 1.5B parameter model), would this be straightforward to do? Could you fine-tune on a subset?**
>
> **A:** We can finetune the model using the full training data. The scaling issue of our approach will come in clustering a dataset of millions/billions tokens. One can consider a random subset of the training data to approximate the latent space. The efficacy of such a solution requires empirical evaluation and is out of the scope of this paper. The current paper aims at providing explanation of a task specific model where generally the training data is not as large as the pretraining dataset.

---

> > ### Comment · Reviewer_6gzN · 2023-11-23
> >
> > Thanks for answering many of my questions. Given the thoughtful answers to the questions, I've increased my score from a 3 to a 5. I still think there's some additional experiments and improvements on the method, including relevant baselines that are important before I can recommend acceptance.

---

### Official Review · Reviewer_jD3R · 2023-11-06

**Soundness:** 2 fair
**Presentation:** 3 good
**Contribution:** 2 fair
**Rating:** 3
**Confidence:** 4

**Summary:**

This paper proposes latent concept attribution method, which works by (1) discovering concepts from a corpus based on hierarchical clustering of representations (2) selecting tokens that has high importance in the sentence (3) mapping selected tokens to extracted concepts with a trained classifier (4) generating natural language explanations with concepts and LLMs. Experiments show that the proposed approach produces plausible explanations.

**Strengths:**

- Explaining LLM predictions with concepts and natural language is an interesting research direction which is beneficial to broader users of NLP systems.
- Break-down evaluation of each component in the proposed method in Sec. 3.3 is useful.

**Weaknesses:**

**In the current state, the most significant weakness of the paper is the experiments.**

- The paper lacks comparison to other explanation algorithms in the experiments.
- The quality of the generated natural language explanation is evaluated with case studies only.

 I understand that evaluation of explanation-based algorithms are tricky, especially for natural language explanations that the authors study. To evaluate utility of explanations, the authors can perform human evaluation. Here I suggest some baselines and ablation studies (1) generating rationales with LLMs (Plausifier in Sec. 2.4) directly based on inputs and predictions (2) skipping the conceptmapper, and directly generating explanations with extracted salient words (in Sec. 2.2) and LLMs. For evaluation of utility, there is a number of references such as [1].

[1] Sun et al. Investigating the Benefits of Free-Form Rationales, 2022

I also hope to see quantitive evaluation of faithfulness, which is crucial for explanation algorithms, but is not quite intuitive for generative explanations. I hope to hear author's thoughts about this.

**The advantage of concept-based explanation to other attribution-based algorithms is not clear from the experiments.**

In introduction, the authors claim that a limitation of attribution methods is because of "multi-facet of words in different contexts". I hope to see the point supported by case studies in experiments.

**Experiments are on sequence tagging and classification tasks only**

I wonder whether concept-based explanations are applicable to more complex reasoning tasks, like reading comprehension and commonsense reasoning. In this case, what explanation outputs do the authors expect?

**Questions:**

Following my points mentioned in the weakness, I hope authors can address:
1. What is the plan for evaluating practical utility and faithfulness of the generated explanations?
2. How can the approach be applied to more complicated NLP tasks such as reading comprehension and commonsense reasoning?

---

> ### Author Response · Authors · 2023-11-21
>
> **W1: The paper lacks comparison to other explanation algorithms in the experiments.**
>
> **W2: The quality of the generated natural language explanation is evaluated with case studies only.**
>
> **A:** The automatic evaluation of the free form explanation is difficult. For this reason, we separately performed automatic evaluation of each component of LACOAT’s pipeline to quantify its correctness. As described in the paper, the sequence labeling task (POS tagging) was deliberately considered in the study to enable a controlled evaluation of each component. We believe that a human evaluation of the explanation will suffice as an evaluation of the system as suggested by the reviewer earlier.
>
> **S: I understand that evaluation of explanation-based algorithms are tricky, especially for natural language explanations that the authors study. To evaluate the utility of explanations, the authors can perform human evaluation. Here I suggest some baselines and ablation studies (1) generating rationales with LLMs (Plausifier in Sec. 2.4) directly based on inputs and predictions (2) skipping the conceptmapper, and directly generating explanations with extracted salient words (in Sec. 2.2) and LLMs. For evaluation of utility, there is a number of references such as [1].**
>
> **A:** Thank you for suggesting various baselines to compare in human evaluation. We will design a study and will perform human evaluation.
>
> **Q: I also hope to see quantitative evaluation of faithfulness, which is crucial for explanation algorithms, but is not quite intuitive for generative explanations. I hope to hear the author's thoughts about this.**
>
> **A:** LACOAT provides explanation both at latent concept level and as free form explanation. As mentioned by the reviewer, quantifying faithfulness in the case of freeform explanation is hard. However, we argue that the last layer latent concept-based explanation is faithful since the main attributed token at the last layer is the classification head and its latent concept is a reflection of how the model relates it with specific knowledge of the training data.
>
> **W/Q: The advantage of concept-based explanation to other attribution-based algorithms is not clear from the experiments. In the introduction, the authors claim that a limitation of attribution methods is because of "multi-facet of words in different contexts". I hope to see the point supported by case studies in experiments.**
>
> **A:** We will include specific examples to support the claim.
>
> **W/Q: Experiments are on sequence tagging and classification tasks only. I wonder whether concept-based explanations are applicable to more complex reasoning tasks, like reading comprehension and commonsense reasoning. In this case, what explanation outputs do the authors expect?**
>
> **Q: How can the approach be applied to more complicated NLP tasks such as reading comprehension and commonsense reasoning?**
>
> **A:** The output of LACOAT varies depending on the task as can be seen from the presented sequence labeling task and the sequence classification task where in the former the latent concepts consist of words while in the latter case they consist of sentences. The Plausifier explanation will also be task dependent. We performed experiments using the MNLI task and found latent concepts at the last layer consisting of sentence pairs. The sentences are grouped together in a latent concept based on how the hypothesis relates with the premise e.g. summary, rephrasing the argument, logical relationship, etc.
>
> **Q: What is the plan for evaluating practical utility and faithfulness of the generated explanations?**
>
> **A:** We aim to include the evaluation of utility in human annotation, where we will explicitly ask the evaluator if the latent concept as well as the explanation add any value to the underlying task. We deem our explanations to be faithful based on the efficacy of the underlying modules (Attribution and Latent Concepts). Given this we argue that our overall explanation is faithful to the model.

---

### Author Response · Authors · 2023-11-21

We thank the reviewers for their insightful comments, questions and suggestions. We include our response below each review. Questions **(Q)**, Weaknesses **(W)** and Suggestions **(S)** are marked in bold, followed by their specific response prefixed with **(A)**.

---

### Meta-Review · Area_Chair_aYMF · 2023-12-02

**Metareview:**

This paper presents a method for explaining predictions based on latent concepts. Rather than attributing a prediction to a token, we can instead attribute the prediction to a particular concept associated with a token, such as the concept "US President" associated with the token "Trump." This process consists of several steps: concept discovery, attribution to words, mapping to concepts, and then "plausifyer" to render the results in a human-understandable form (using ChatGPT).

While the ideas in this paper are interesting, the reviewers highlight several weaknesses. Most critically, the evaluation is thin: the method is not well situated with respect to other methods from the literature, and the evaluation of the final language explanations is qualitative only. The evaluations of ConceptDiscoverer and PredictionAttributor are somewhat artificial and rely on the simple nature of the tasks involved. Some reviewers also critiqued this choice of tasks.

The paper would likely be stronger if a future version can use human evaluation as suggested by jD3R; experiments on simulatability could be used here to compare this format of explanations to others from the literature.

**Justification For Why Not Higher Score:**

No reviewer recommends acceptance and major issues with the paper's evaluation are raised.

**Justification For Why Not Lower Score:**

N/A

---

### Decision · Program_Chairs · 2024-01-16

Reject